# How Does Environmental Regulation Affect the Relationship between FDI and Technological Innovation: From the Perspective of Technology Transactions

**Meng Zeng [1], Lihang Liu [1], Fangyi Zhou [2] and Yigui Xiao [3,4,***

[1] School of Public Administration, Central South University, Changsha 410083, China; 20140058@csuft.edu.cn (M.Z.); lihangliu@csu.edu.cn (L.L.)
[2] Business School, Central South University, Changsha 410083, China; zhoufangyi@csu.edu.cn
[3] College of Business, Hunan University of Technology, Zhuzhou 412007, China
[4] School of Business Administration, Hunan University of Finance and Economics, Changsha 410205, China
[*] Correspondence: csu_zm@csu.edu.cn

**Abstract:** Many studies have found that FDI can reduce the pollutant emissions of host countries. At the same time, the intensity of environmental regulation would affect the emission reduction effect of FDI in the host country. This study aims to reveal the internal mechanisms of this effect. Specifically, this paper studies the impact of FDI on technological innovation in China's industrial sectors from the perspective of technology transactions from 2001 to 2019, and then analyzes whether the intensity of environmental regulation can promote the relationship. Results indicate that FDI promotes technological innovation through technology transactions. In addition, it finds that the intensity of environmental regulation significantly positively moderates the relationship between FDI and technological innovation, which is achieved by positively moderating the FDI–technology transaction relationship. Regional heterogeneity analysis is further conducted, and results show that in the eastern and western regions of China, FDI can stimulate technological innovation within regional industrial sectors through technology trading. Moreover, environmental regulation has a significant positive regulatory effect on the above relationship, but these effects are not supported by evidence in the central region of China.

**Keywords:** FDI; innovation; technology transactions; environmental regulation; heterogeneity analysis

## 1. Introduction

China, which is the largest developing country in the world, has attracted tremendous foreign direct investment (FDI) since the reform and opening-up policy [1]. FDI is an important source of access to technology and resources for some developing countries as foreign firm entry is accompanied by technology and knowledge transfer into the host economy [2]. However, there are also many environmental issues caused by FDI from transnational corporations (TNCs) that aim to decrease labor costs and evade stringent environmental regulations in their home countries [3]. FDI embedded with high pollution and new technology brings in high-speed economic growth as well as huge environmental pollution in China. It was considered as one of the most important factors causing heavy pollution in China because developed countries have treated China as the "Pollution Haven" by reason of lower environmental standards [4]. In recent decades, the Chinese government has designed and implemented strict environmental regulations aiming at receiving FDI with greener technology for driving technological progress and dealing with environmental problems in China [5]. Some studies indicate that high-quality FDIs may enhance the total technological level of host countries due to technology spillover [6].

Hence, in order to not violate the laws of China, TNCs have to enter China with more high-quality FDIs that may incur innovations of domestic producers by technology

spillover from TNCs with the improvement of environmental regulation intensity [7]. Producers can improve productivity through technological innovation and reduce the negative impact of production behaviors on the environment under external environmental pressure [8]. Environmental regulations entail satisfying environmental protection requirements as soon as possible to avoid administrative punishment and reduce the environmental costs [9]. Technology purchases in technology markets can not only quickly achieve environmental compliance, but also reduce the uncertainty and risk of innovation for some producers. Moreover, producers can also redevelop the purchased technology through technology absorption and finally achieve environmental technology improvement [10]. FDI is an important source of technology for producers in the host country as they can improve their productivity by purchasing advanced technologies embedded in FDI when environmental regulations encourage high-quality FDI to enter the host country. Therefore, this study aims to reveal (1) whether FDI stimulates technological innovation in China's industrial sectors and whether this relationship is affected by the intensity of environmental regulation, (2) whether technology transactions play a role in the FDI–technological innovation relationship, and (3) whether there exist regional differences in these impacts mentioned above in China.

This study employs data from 30 provinces of China from 2001 to 2019 and investigates whether FDI has an impact on technological innovation through technology transactions. Furthermore, this study examines the impact of environmental regulation on the relationship between FDI, technology transactions, and technological innovation. Results reveal that FDI can trigger the technological innovation of industrial sectors in China through stimulating technology transactions. We further find out that intensity of environmental regulation exerts a positive moderating effect both on the relationship between FDI and the technological innovation of industrial sectors, and on the relationship between FDI and technology transactions. Finally, results of the heterogeneity study based on different regions in China indicate that FDI exerts a positive impact on technological innovation and technology transactions in the eastern and western regions, but not in the central region. In the eastern and western regions, the intensity of environmental regulation has a positive impact both on the FDI–technological innovation relationship and on the FDI–technology transaction relationship. There is no evidence of such an effect in the central region. Literature contributions of this study are as follows. First, in the host country, technology transactions are an important and effective path of FDI on technology innovation. In previous studies, the positive effect of FDI on innovation are mainly realized through imitation, absorption, and re-innovation [11,12]. However, technology absorption and re-innovation require a longer period, and may encounter higher risk of capital and failure at the same time [13]. Producers can acquire the advanced technology embedded in FDI through market transaction, which can shorten the R&D cycle, reduce the risk of R&D, and decrease the need for finance. Second, we find that environmental regulation has become a crucial driving force for the spillover and transformation of technology embedded in FDI through stimulating technology transactions in China. When the intensity of environmental regulation increases in the host country, TNCs have to offer high-quality FDI that benefits local producers to meet the local environmental protection requirements. Producers can quickly access advanced technology to optimize environmental cost from high-quality FDI through technology transactions and reduce the risk of environmental penalties when the government raises environmental requirements [14,15]. Therefore, environmental regulation strengthens the technology spillover effect from FDI and stimulates technology transactions between local and multinational enterprises in China.

This paper is structured as follows. Literature review and hypothesis development are in Section 2. Section 3 presents research design that includes data, samples, variables, and model construction. Empirical results, analysis, and robustness tests are reported in Section 4. In Section 5, conclusions and implications are discussed.

## 2. Conceptual Model and Literature Review

Important drivers of outbound investment are labor costs, raw material costs, and domestic institutional pressures for TNCs [16,17]. Additionally, FDI is an important means to drive domestic economic development, especially for developing countries [18]. However, most FDIs transferred by TNCs are concerned with low-technology and high-pollution businesses because of the imperfect institutions in developing countries. Although developing countries have achieved economic development by attracting FDIs, they have also become a pollution haven for developed countries and TNCs [19]. Since the reform and opening-up, China has received a large number of low-end FDIs from developed countries and promoted its own economic development [20].

However, it has also brought about serious environmental pollution [21]. Moreover, from the dual perspective of technology and economic development, the real driving effect of FDI on China's technology is extremely limited, and the economic effect is overestimated [22]. With China's emphasis on environmental issues, our environmental protection has gradually improved, and production activities have been subject to strict environmental review. For instance, FDIs concerned with high pollution, and low technologies are not allowed to flow into China [23]. Therefore, the spillover effect of high-quality FDIs from developed countries on China's environmental innovation may be affected by the intensity of environmental regulation. In addition, as the intensity of environmental regulation is on the rise, the impact path of FDI on the host country's environmental innovation may also change, which could be different from the previous learning and imitation effects through technology spillovers, and more likely from the technology transactions between different producers. We also find some different results in other developing countries. Haug and Ucal [24] presents new evidence that FDI has no statistically significant impact on environmental performance in Turkey. Results by Bakhsh, et al. [25] confirm that, in Pakistan, FDI can increase the economic level and that a high economic level leads to lower pollutant emissions. Rafique, et al. [26] examined the relationship between FDI and technological innovation in BRICs countries and found no positive impact of FDI on technological innovation.

A conceptual model which involves three relationships is presented in Figure 1. Specifically, it is expounded in our model whether environmental regulation moderates the relationship between FDI and industrial innovation, and if there is a significant impact of FDI on industrial innovation. Second, it states whether technology transactions have a mediating effect on the relationship between FDI and industrial innovation. Third, whether environmental regulation moderates the relationship between FDI and technology transactions, presuming the mediating effect exists, is elaborated.

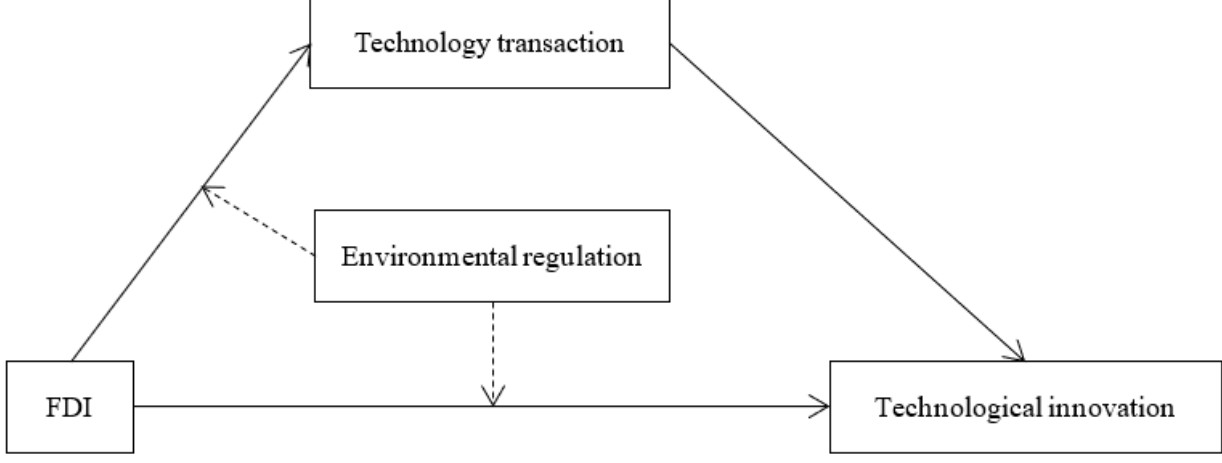

**Figure 1.** Conceptual model.

Knowledge is usually transferred through the cooperation of different individuals among industries. Knowledge transfer can promote the integration of explicit and tacit knowledge from technologies of TNCs [27]. In addition, knowledge transfer is conducive to the learning and imitation of new technologies, knowledge, and experience of TNCs [28]. Long-term cooperation between regional enterprises and TNCs enables the latter to get highly involved in the production, design, and development of host country enterprises, accelerating knowledge flows between TNCs and host country enterprises [29]. Moreover, because of the deeply embedded global value chain, barriers of technical information flow to host enterprises of the global value chain are broken, and the cost of acquiring external knowledge will be greatly reduced by host enterprises [30].

Human capital flows boost the interaction and collaboration in technological innovation between regional enterprises and R&D institutions of TNCs through employee education, skills and knowledge training, and experience cultivation [31,32]. The interaction of human capital improves communication and deep communication frequency and strengthens the trust level between the two sides. The effect of personnel mobility further improves the openness of multinational and regional enterprises, forms a platform for information sharing and knowledge transmission, and improves the speed of innovation and the value of enterprises [33,34]. FDI of TNCs can also ease the constraints of innovation financing for regional enterprises, lower the high innovation cost caused by the low financial efficiency in the host country, and provide financial support for R&D of regional enterprises [35]. TNCs deeply participate in the innovation activities of regional companies and become stakeholders through financing activities. With the deepening of regional corporate innovation activities, TNCs can also provide assistance for host country enterprises by updating equipment, recruiting personnel, and transferring technology [36].

FDI has had a crucial impact on the environment of the host country. Due to the lag in the environmental system of the host country, FDI often exacerbates environmental pollution [37]. Moreover, the lagging institutional environment of host countries makes TNCs regard developing countries as a "Pollution Haven" [38]. As the Chinese government attaches greater importance to environmental protection, intensity of environmental regulation in China is on the rise. The Chinese governments tend to "screen" high-quality FDIs for greener economy benefits and sustainable development through strict environmental regulations [39]. High-quality FDIs concerned with advanced technologies could boost China's energy efficiency and total factor productivity, and reduce pollution emissions [40]. Wang and Luo [41] found that when the level of FDI is low, the capacity for scientific and technological innovation aggravates environmental pollution levels, while when the level of FDI crosses a higher threshold, the capacity for scientific and technological innovation improves environmental quality. Hao, et al. [42] also confirm the existence of the "pollution halo hypothesis", which means that technological innovation can reduce pollutant emissions. Domestic firms and industries can increase further innovation on the basis of absorbing advanced technology. There are not only positive impacts of environmental regulation on the quality of FDI, but also on the innovation of domestic industries. For domestic firms, strict environmental regulation can trigger firms' innovation that offset the innovation costs and promote competitiveness [8].

As TNCs are deeply embedded in the global value chain, barriers to the flow of technical information across the global value chain to host country enterprises are broken, and the cost of acquiring external knowledge for host country enterprises is significantly reduced [30]. Host country enterprises can obtain advanced technology from TNCs through technology transfer or licensing [36]. For many enterprises, access to technology transfer or licensing can reduce the cost of research and development, and more significantly trigger technology spillover effect of FDI in a short period of time [43]. Technology transfer through host country enterprises not only produces technology spillover effects within the industry, but also generates the transfer of new technologies and knowledge across value chains and industries [44]. For host country enterprises, the technology spillover effect from domestic enterprises removes barriers to access to technology due to intellectual property

protection and institutional differences at home and abroad, as well as geographical barriers to technology flows [45]. Technology spillovers from domestic enterprises are easier to imitate and learn from intra-industry enterprises, and enterprises within and between industries need only lower costs to complete the acquisition of technologies compared to the past [46,47]. Therefore, for industries of host countries, FDI can promote industry innovation through stimulating technology trading which could enable the diffusion of knowledge and technology within and between industries.

With the increase in the intensity of environmental regulations, TNCs have to improve the quality of FDI to meet the environmental standards of the host country. High-quality FDI concerned with advanced technology could promote China's energy efficiency and total factor productivity and reduce pollution emissions [40]. For domestic enterprises and industries, increase in the intensity of environmental regulation requires enterprises to upgrade their technology as soon as possible to meet the increasingly stringent environmental standards to avoid penalties. In response, it is an appropriate strategy to quickly acquire new technologies from the market, especially for non-advanced technology enterprises in the host country [10,48]. As mentioned earlier, buying technology in the market can shorten the time it takes to develop technology and meet environmental requirements in a short period of time, which is more obvious and less costly than independent innovation [49]. With the advanced technology brought about by FDI from the market, domestic companies and industries will carry out R&D and innovation based on the technology they have acquired. FDI can promote technological innovation in host countries by stimulating technology transactions to spread knowledge and technology within and between industries. With the increase in environmental regulation intensity, TNCs have to improve the quality of FDI to meet the environmental standards of the host country.

## 3. Sample, Data and Variables

### 3.1. Sample and Data

In this study, industrial sectors of 30 provinces and regions in China are selected as research samples. Tibet, Hong Kong, Macau, and Taiwan are excluded in this study as data are not available. Industrial sectors, in this study, include the manufacturing industry, mineral mining industry, and energy production industry. Innovation data come from the Yearbook of Industrial Science and Technology Statistics. Environmental regulation data come from the Yearbook of China's Environmental Statistics and the environmental statistics bulletin of various provinces and cities. Industry related data come from the Yearbook of China's Industrial Statistics, and data of marketization and urbanization come from China City Statistical Yearbook. The interpolation method is used to supplement the missing data. At the same time, this study uses the 1990 price index as the base period to deflate economic indicators to eliminate the impact of price changes on the results.

### 3.2. Variables

FDI, measured by amount of FDI received by industrial sectors each year [50], is an independent variable in this study. Technological innovation is employed as a dependent variable which is measured by the number of patent of industrial sectors [51]. Technology transactions is the mediating variable and measured by volume of technology transactions [52]. Environmental regulation is employed as the moderating variable and measured by investment in environmental governance according to Yuan and Xiang [53]. In this study, investment in environmental governance means expenditure from industrial sectors for reducing pollution from wastewater, waste gas and solid waste produced by industrial sectors. We also choose R&D capability measured by full-time equivalent of R&D [54], inventory [55], profit [56], fixed investment [57], size measured by number of employee [58], urbanization [59], and marketization [60] as controls in this study.

## 4. Empirical Results

### 4.1. Descriptive Statistics

Descriptive statistics in Table 1 mainly include mean, maximum, minimum, and standard deviation. Results show that the value of a variable has significant difference and obvious variation, which means the unbalanced development of different regions in China. For several major variables, the maximum of FDI is 2500 times the minimum. The maximum value of innovation (I) is 272,616, which is 9000 times the minimum. The minimum value of technology transactions (TT) is only 6 million RMB, which indicates a lower level of technology circulation in this region. In statistics of other variables, large numerical differences exist across regions.

**Table 1.** Descriptive statistics.

| Variable | n | Mean | Max | Min | Sd |
|---|---|---|---|---|---|
| foreign direct investment (FDI) | 570 | 1106 | 19,533 | 7 | 2091 |
| innovation (I) | 570 | 13,437 | 272,616 | 31 | 28,072 |
| technology transactions (TT) | 570 | 203.71 | 5695.28 | 0.06 | 536.12 |
| environmental regulation (ER) | 570 | 70.45 | 1169 | 0.79 | 117.7 |
| rd_human (RDH) | 570 | 555.7 | 6425 | 3 | 908.0 |
| inventory (INV) | 570 | 22.89 | 170.2 | 0.330 | 27.09 |
| profit (P) | 570 | 15.07 | 123.9 | −1.500 | 20.16 |
| fixed investment (FI) | 570 | 100.3 | 615.1 | 1.580 | 113.6 |
| employment (E) | 570 | 6.780 | 24.51 | 0.310 | 6.050 |
| urbanization (U) | 570 | 48.97 | 89.60 | 11.54 | 17.04 |
| marketization (M) | 570 | 5.950 | 10.33 | 2.330 | 1.760 |

Table 2 shows the correlation of variables in this study. FDI is highly correlated with innovation (I), technology transactions (TT), and environmental regulation (ER), respectively, which may indicate a high degree of linear relationship. Technology transactions (TT), environmental regulation (ER), urban (U), and marketization (M) are highly correlated with innovation indicating that these variables may have important impacts on innovation (I). Environmental regulation is also significantly correlated with RDH, urbanization (U) and marketization (M), respectively.

**Table 2.** Correlation matrix.

| | FDI | I | TT | ER | RDH | INV | P | FI | E | U | M |
|---|---|---|---|---|---|---|---|---|---|---|---|
| FDI | 1 | | | | | | | | | | |
| I | 0.87 ** | 1 | | | | | | | | | |
| TT | 0.93 *** | 0.87 * | 1 | | | | | | | | |
| ER | 0.91 *** | 0.90 ** | 0.91 * | 1 | | | | | | | |
| RDH | 0.81 * | 0.95 | 0.81 *** | 0.82 *** | 1 | | | | | | |
| INV | 0.82 | 0.90 * | 0.82 | 0.81 | 0.95 * | 1 | | | | | |
| P | 0.73 * | 0.82 ** | 0.73 ** | 0.74 | 0.89 *** | 0.93 ** | 1 | | | | |
| FI | 0.51 | 0.67 | 0.51 | 0.62 | 0.71 | 0.78 *** | 0.85 * | 1 | | | |
| E | 0.53 | 0.69 | 0.53 | 0.54 | 0.79 | 0.82 | 0.81 * | 0.70 ** | 1 | | |
| U | 0.50 | 0.38 ** | 0.50 *** | 0.52 * | 0.38 * | 0.46 ** | 0.41 *** | 0.38 *** | 0.15 | 1 | |
| M | 0.80 *** | 0.89 *** | 0.80 | 0.85 *** | 0.92 *** | 0.95 *** | 0.94 | 0.87 *** | 0.81 | 0.46 ** | 1 |

Standard errors in parentheses * $p < 0.05$, ** $p < 0.01$, *** $p < 0.001$.

### 4.2. Main Effect and Mediating Effect

Table 3 displays the empirical results on the main effect and mediating effect in this study. Columns 1 and 3 report empirical results on the impact of FDI on innovation without control variables based on fixed-effect regression and Poisson regression. Columns 2 and 4 report empirical results on the impact of FDI on innovation including all variables based on fixed-effect regression and Poisson regression. Columns 5 and 6 report results on the

mediating effect based on the method proposed by Baron and Kenny [61], and the last two lines in Table 3 show the testing results on the mediating effect according to Sobel [62]. Individual effect and year effect are also controlled.

**Table 3.** Regression on the main effect and mediating effect.

| | Innovation-Fe | | Innovation-Poisson | | Mediating Effect | |
|---|---|---|---|---|---|---|
| | Model-1 | Model-2 | Model-3 | Model-4 | Model-5 | Model-6 |
| FDI | 4.25 *** | 4.18 *** | 2.762 *** | 1.88 *** | 4.42 ** | 3.99 *** |
| | (0.35) | (0.38) | (0.00) | (3.54) | (0.22) | (0.37) |
| rd_hum | | 8.95 *** | | 0.58 *** | 4.75 *** | 6.79 *** |
| | | (1.04) | | (1.28) | (1.03) | (1.26) |
| inventory | | 1.41 | | 0. 51 * | −9.22 ** | 9.85 |
| | | (5.10) | | (2.45) | (3.45) | (5.43) |
| profit | | 2.24 * | | 0.92 *** | 6.75 * | 9.35 ** |
| | | (1.66) | | (3.04) | (3.01) | (5.10) |
| fixed inv | | 3.85 | | 0.30 * | −3.344 *** | 23.62 ** |
| | | (7.85) | | (1.65) | (0.46) | (7.96) |
| employment | | 13.3 | | 1.61 *** | 14.97 ** | 17.47 |
| | | (8.84) | | (9.52) | (16.85) | (18.67) |
| urban | | 3.18 ** | | 4.21 *** | 3.48 ** | 3.92 ** |
| | | (7.36) | | (7.83) | (4.40) | (6.77) |
| marketization | | 29.58 ** | | 1.38 *** | 6.04 ** | 2.18 *** |
| | | (12.09) | | (2.02) | (10.70) | (12.46) |
| tech_trans | | | | | | 4.54 ** |
| | | | | | | (0.73) |
| _cons | 14.56 *** | 29.48 *** | 7.36 *** | 6.97 *** | 31.20 | 20.61 *** |
| | (8.57) | (14.73) | (2.15) | (1.83) | (9.18) | (9.93) |
| $R^2$ | 0.86 | 0.93 | | | 0.83 | 0.92 |
| indv-fixed | yes | yes | yes | yes | yes | yes |
| year-fixed | yes | yes | yes | yes | yes | yes |
| N | 570 | 570 | 570 | 570 | 570 | 570 |

| Sobel test | Coef | Std Err | Z | $p > |Z|$ |
|---|---|---|---|---|
| | 0.38 | 0.11 | 3.54 | 0.0004 |

Standard errors in parentheses * $p < 0.05$, ** $p < 0.01$, *** $p < 0.001$.

Empirical results show that FDI has a significant positive impact on innovation (coefficient = 4.18, $p < 0.001$; coefficient = 1.88, $p < 0.001$). In the test of mediating effect, FDI turns out to have a significant positive impact on technology transactions (coefficient = 4.42, $p < 0.01$). After adding tech_trans into Model-6, FDI still has a significant positive impact on innovation (coefficient = 3.99, $p < 0.001$) and the mediating variable (tech_trans) also has an impact on innovation (coefficient = 4.54, $p < 0.01$). The Sobel test (Z = 3.54, $p > 0.0004$) supports the mediating effect of technology transactions (tech_trans) on the relationship between FDI and innovation. Therefore, we consider that FDI promotes innovation by stimulating technology transactions.

From results of the controls, R&D capability (rd_hum) influences the technological innovation of the regional industrial sector. This is because the stronger the R&D capability, the more patent-based technological innovation output in the region. Profit has a significant positive impact on technological innovation in China's industrial sectors across regions. For the industrial sector in a region, sustained profitability can provide a stable financial guarantee for technology R&D, and it has a strong ability to resist the uncertainty of R&D risks. Urbanization and marketization have also promoted technological innovation in the regional industrial sector as they can accelerate the flow of information and resources among different innovation entities, and play a significant role in the optimal allocation of resources in a region. Inventory, fixed investment, and industrial scale turn out to have no significant impact on technological innovation in the regional industrial sector in this study.

### 4.3. Robustness Test

Table 4 shows robustness tests of the main effect regression. For the robustness test, two methods are used in this paper: the adjustment time window and replacement variables. Regression results in the first column and the second column are based on the time window of 2005–2012 and the time window of 2011–2019, respectively. Regression results in the third column are based on the use of R&D expenditure as the proxy variable for technological innovation [63]. Regression results in the fourth column are based on the use of invention patent grants as the proxy variable for technological innovation [64]. Regression results based on the adjusted time window and replacement variables support the main effect; that is, FDI has a significant positive impact on technological innovation, which also shows that the regression results of Table 3 are robust.

**Table 4.** Robustness test.

|  | Adjusting Time | | Alternative Variable | |
|---|---|---|---|---|
|  | **2005–2012** | **2011–2019** | **RD Expenditure** | **Invention** |
| FDI | 2.82 ** | 4.38 *** | 0.95 ** | 8.34 *** |
|  | (1.11) | (0.47) | (0.16) | (0.95) |
| rd_hum | 13.25 *** | 13.60 *** | 0.17 *** | 7.15 *** |
|  | (1.22) | (1.77) | (0.09) | (1.58) |
| inventory | 3.06 | 4.97 * | 6.63 * | 1.07 |
|  | (8.54) | (5.53) | (0.48) | (6.52) |
| profit | 2.04 ** | 3.03 ** | 1.14 *** | 2.03 ** |
|  | (1.04) | (1.48) | (2.42) | (2.89) |
| fixed investment | 3.56 | 2.15 * | 0.33 ** | 1.53 |
|  | (14.59) | (12.39) | (0.26) | (4.46) |
| employment | 11.19 * | 9.18 | 34.56 | 22.03 ** |
|  | (6.39) | (6.03) | (12.35) | (16.13) |
| urbanization | 4.69 *** | 3.18 *** | 1.50 ** | 3.62 |
|  | (6.09) | (7.63) | (0.61) | (1.84) |
| marketization | 40.85 * | 19.13 ** | 0.72 *** | 1.48 ** |
|  | (19.79) | (20.59) | (0.85) | (2.26) |
| _cons | 28.37 *** | 20.05 ** | 23.55 | −22.28 ** |
|  | (9.46) | (6.36) | (7.98) | (8.59) |
| $R^2$ | 0.92 | 0.89 | 0.91 | 0.93 |
| indv-fixed | yes | yes | yes | yes |
| year-fixed | yes | yes | yes | yes |
| N | 240 | 270 | 570 | 570 |

Standard errors in parentheses * $p < 0.05$, ** $p < 0.01$, *** $p < 0.001$.

From the results of the controls, results about R&D capability (rd_hum) in the robustness test are consistent with the results of the main effect, which indicates that R&D capability has a significant positive impact on the technological innovation of the regional industrial sector. Profit has a significant positive impact on technological innovation in various regions of China's industrial sectors, which also proves that only sustained profitability can have a strong support effect on technological innovation. In the robustness test, urbanization does not exert a significant impact on technological innovation, which is different from the results of the main effect. Thus, it is unable to prove that urbanization is an important driving factor for innovation. Marketization also has a significant positive effect on technological innovation in the robustness test, which shows that marketization is a crucial driving force of technological innovation by optimizing resource allocation.

### 4.4. Moderating Effect of Environmental Regulation

Table 5 reports the moderating effects of environmental regulation on the FDI–innovation relationship and on the FDI–technology transaction relationship. Columns 1 and 2 display the moderating effect of environmental regulation on the relationship between FDI and innovation. Results reveal that environmental regulation has a positive moderating effect

on the relationship between FDI and innovation (coefficient = 0.71, $p < 0.001$) based on the result that FDI has a positive effect on innovation (coefficient = 4.18, $p < 0.001$). Columns 3 and 4 report the moderating effect of environmental regulation on the relationship between FDI and technology transactions. We also find that environmental regulation has a positive moderating effect on the relationship between FDI and technology transactions (coefficient = 1.08, $p < 0.01$).

**Table 5.** Moderating effect of environmental regulation.

| | Innovation | | Tech_Trans | |
|---|---|---|---|---|
| FDI | 4.18 *** | 0.93 * | 4.42 ** | 4.07 ** |
| | (0.38) | (0.44) | (0.22) | (0.13) |
| env regulation | 66.63 *** | 47.53 *** | 2.59 *** | 2.88 *** |
| | (6.64) | (6.10) | (0.39) | (0.40) |
| rd_hum | 8.95 *** | 6.38 *** | 4.75 *** | 4.36 *** |
| | (1.04) | (0.95) | (1.03) | (0.86) |
| inventory | 1.41 | 1.37 *** | 9.22 ** | 9.94 ** |
| | (5.10) | (5.25) | (3.45) | (3.43) |
| profit | 2.24 ** | 1.58 *** | 6.75 * | 5.91 * |
| | (1.66) | (1.45) | (3.01) | (3.01) |
| fixed inv | 3.85 | 6.20 * | 3.34 ** | 3.00 |
| | (7.85) | (7.21) | (0.46) | (0.47) |
| employment | 13.31 | 12.09 | 14.97 ** | 16.74 ** |
| | (8.84) | (6.14) | (6.85) | (5.70) |
| urbanization | 3.18 ** | 2.87 *** | 3.48 ** | 3.03 *** |
| | (7.36) | (6.83) | (4.40) | (4.37) |
| marketization | 29.58 ** | 30.05 ** | 6.04 ** | 5.51 *** |
| | (12.09) | (11.05) | (10.70) | (0.72) |
| FDI * env regulation | | 0.71 *** | | 1.08 ** |
| | | (0.23) | | (0.62) |
| _cons | 29.48 *** | 25.57 ** | 31.20 | 39.34 *** |
| | (14.73) | (15.42) | (9.18) | (7.56) |
| $R^2$ | 0.93 | 0.82 | 0.83 | 0.81 |
| indv-fixed | yes | yes | yes | yes |
| year-fixed | yes | yes | yes | yes |
| N | 570 | 570 | 570 | 570 |

Standard errors in parentheses, * $p < 0.05$, ** $p < 0.01$, *** $p < 0.001$.

*4.5. Heterogeneity Analysis*

Due to the huge development differences in economic development and geographic position across various regions in China, this study has divided samples into various provinces and cities in China. Based on previous research by Yuan and Xiang [53] and Yang and Li [50], this study divides China into three regions which include Eastern China, Central China, and Western China, in which relationships among regional FDI, innovation, technology transactions, and environmental regulation are explored. Eastern China includes Beijing, Tianjin, Hebei, Liaoning, Shanghai, Jiangsu, Zhejiang, Fujian, Shandong, Guangdong Guangxi, and Hainan. Central China includes Shanxi, Henan, Inner Mongolia, Anhui, Hubei, Hunan, Jiangxi, Jilin, and Heilongjiang. Western China is Shaanxi, Gansu, Ningxia, Qinghai, Xijiang, Chongqing, Sichuan, Guizhou, and Yunnan.

Table 6 reports heterogeneity analysis on the main effect and mediating effect. Columns 1–3 report the main effect and mediating effect in the eastern region, columns 4–6 report the main effect and mediating effect in the central region, and columns 7, 8, and 9 report the main effect of the central region and the mediating effect in the central region. Results indicate that in the eastern and western regions, FDI stimulates technological innovation (coefficient = 5.42, $p < 0.001$; coefficient = 2.24, $p < 0.01$), which is achieved by stimulating technology exchanges (coefficient = 3.13, $p < 0.01$; coefficient = 2.87, $p < 0.001$). However, the positive stimulus effect of FDI on technology has not been realized in the central region.

**Table 6.** Heterogeneity analysis on the main effect and mediating effect.

| | Eastern Region | | | Central Region | | | Western Region | | |
|---|---|---|---|---|---|---|---|---|---|
| FDI | 5.42 *** | 1.97 * | 5.18 *** | 1.39 | 0.75 | 2.52 | 2.24 * | 0.85 *** | 2.63 * |
| | (0.58) | (0.04) | (0.57) | (1.99) | (0.06) | (1.93) | (0.98) | (0.08) | (1.11) |
| rd_hum | 16.65 *** | 0.55 *** | 15.07 *** | 28.11 *** | 0.73 *** | 20.60 *** | 8.72 ** | 0.82 ** | 7.42 * |
| | (1.53) | (0.10) | (1.59) | (3.51) | (0.11) | (3.76) | (3.26) | (0.26) | (3.38) |
| inventory | 22.40 | 21.38 ** | 14.54 | 10.62 | 21.53 *** | 12.36 | 15.29 | 18.85 | 15.37 |
| | (10.54) | (6.89) | (10.04) | (14.97) | (4.95) | (15.12) | (9.46) | (7.22) | (9.82) |
| profit | 2.90 ** | 0.91 ** | 2.59 ** | 1.55 | 0.91 | 2.33 | 2.61 ** | 0.30 *** | 3.04 ** |
| | (9.30) | (6.07) | (8.14) | (6.38) | (2.12) | (5.17) | (7.28) | (2.63) | (6.44) |
| fixed inv | 30.90 | 3.45 | 20.07 | 30.09 * | 1.37 ** | 15.71 ** | 35.13 * | 5.667 *** | 33.50 * |
| | (12.49) | (0.84) | (12.78) | (13.68) | (0.44) | (13.35) | (14.57) | (1.18) | (15.82) |
| employment | 26.79 | 20.89 | -33.33 | 27.31 ** | 30.51 | 35.49 ** | 15.59 | 27.77 | 13.93 |
| | (50.91) | (33.65) | (41.50) | (67.07) | (41.73) | (46.45) | (41.25) | (30.21) | (40.27) |
| urbanization | 8.02 | 2.38 | 7.57 | 9.51 *** | 1.70 *** | 5.75 * | 7.77 *** | 2.62 * | 8.73 *** |
| | (12.69) | (10.93) | (15.58) | (19.55) | (6.50) | (21.79) | (16.60) | (4.57) | (16.86) |
| marketization | 38.37 *** | 7.05 *** | 8.58 ** | 9.46 | 5.18 | 6.19 | 25.47 | 3.41 | 8.49 |
| | (21.00) | (1.41) | (9.92) | (18.79) | (0.60) | (19.41) | (18.28) | (1.48) | (18.70) |
| tech_trans | | | 3.13 ** | | | −10.67 | | | 2.87 *** |
| | | | (1.06) | | | (2.54) | | | (1.07) |
| _cons | 29.91 *** | 30.15 | 18.46 *** | 18.27 *** | 19.90 ** | 13.92 | 19.25 | 20.23 | 18.16 |
| | (13.43) | (14.19) | (10.01) | (14.67) | (15.02) | (14.50) | (16.00) | (15.30) | (13.83) |
| $R^2$ | 0.98 | 0.82 | 0.98 | 0.95 | 0.85 | 0.96 | 0.95 | 0.81 | 0.95 |
| indv-fixed | yes | yes | yes | yes | yes | yes | yes | yes | yes |
| year-fixed | yes | yes | yes | yes | yes | yes | yes | yes | yes |
| N | 228 | 228 | 228 | 171 | 171 | 171 | 171 | 171 | 171 |

Standard errors in parentheses, * $p < 0.05$, ** $p < 0.01$, *** $p < 0.001$.

Table 7 reports heterogeneity analysis on the moderating effect of environmental regulation controlling for individual effect and year effect in the empirical results. Table 7 only reports results in the eastern and western regions as FDI has no significant impact on innovation in the central region. Environmental regulation positively moderates the relationship between FDI and innovation in the eastern and western regions (coefficient = 0.65, $p < 0.001$; coefficient = 0.52, $p < 0.01$), and this effect is achieved by positively regulating the relationship between FDI and technology transactions.

**Table 7.** Heterogeneity analysis on the moderating effect of environmental regulation.

| | Eastern Region | | Western Region | |
|---|---|---|---|---|
| | Innovation | Tech_Trans | Innovation | Tech_Trans |
| FDI | 2.50 *** | 0.88 ** | 1.53 ** | 0.66 ** |
| | (0.63) | (0.05) | (1.53) | (0.12) |
| rd_hum | 14.65 *** | 8.97 *** | 10.35 ** | 6.53 * |
| | (1.36) | (1.10) | (1.27) | (1.06) |
| inventory | 1.22 | 1.34 ** | 1.40 | 1.37 |
| | (4.61) | (6.91) | (8.30) | (7.02) |
| profit | 2.68 ** | 8.68 ** | 3.05 *** | 10.72 *** |
| | (1.33) | (5.11) | (1.47) | (6.49) |
| fixed inv | 7.95 | 3.37 *** | 2.46 | 1.09 *** |
| | (1.31) | (0.87) | (1.86) | (1.18) |
| employment | 14.30 | −21.54 | 12.29 | −36.39 |
| | (8.25) | (10.77) | (5.64) | (12.98) |
| urbanization | 5.82 | 4.95 | 3.32 *** | 2.45 *** |
| | (10.93) | (11.02) | (7.95) | (6.61) |

**Table 7.** *Cont.*

|  | Eastern Region | | Western Region | |
|---|---|---|---|---|
|  | Innovation | Tech_Trans | Innovation | Tech_Trans |
| marketization | 22.59 *** | 6.96 *** | 17.94 | 4.21 |
|  | (18.88) | (1.46) | (18.32) | (1.46) |
| FDI * env regulation | 0.65 *** | 0.41 *** | 0.52 ** | 0.36 ** |
|  | (0.03) | (0.02) | (0.01) | (0.01) |
| _cons | 26.67 *** | 20.02 * | 28.78 * | 26.63 |
|  | (13.17) | (12.97) | (15.42) | (11.50) |
| $R^2$ | 0.92 | 0.88 | 0.89 | 0.83 |
| indv-fixed | yes | yes | yes | yes |
| year-fixed | yes | yes | yes | yes |
| N | 228 | 228 | 171 | 171 |

Standard errors in parentheses, * $p < 0.05$, ** $p < 0.01$, *** $p < 0.001$.

In terms of the moderating effect, environmental regulation positively moderates the relationship between FDI and technological innovation in the eastern and western regions, and the effect is achieved by positively moderating the relationship between FDI and technology transactions (coefficient = 0.41, $p < 0.001$; coefficient = 0.36, $p < 0.01$).

## 5. Conclusions and Discussion

Based on the data of Ministry of Industry of 30 provinces in China from 2001 to 2019, this paper explores the impact of FDI on technological innovation and the intermediary mechanism of technology transactions. On this basis, it further analyzes the impact of environmental regulation on the above relationships. Results are as follows.

First, FDI stimulates technological innovation within China's industrial sectors, which is achieved by stimulating regional technology trading. Results of some controls can provide explanation and support for this conclusion. In addition, marketization, which is an important means to optimize the allocation of resources, has been showing a significant positive impact on technology innovation and technology trading. With the improvement of marketization of various regions in China, the technology spillover effect contained in FDI can rapidly diffuse in the market and be absorbed by means of technology transfer. Therefore, marketization has a positive effect on innovation by promoting the technology trade among production departments. At the same time, the study concludes that profit and R&D capability are also important factors. High profits obtained by the production sector provide funds to purchase and absorb the advanced technology brought by FDI through technology transactions, so as to enhance the innovation level of the sector. R&D capability ensures the absorption and redevelopment of new technologies acquired through technology transactions, and further forms the technical innovation of the department.

Secondly, environmental regulation has a significant positive impact on the FDI–technological innovation relationship, which is achieved by positively stimulating the relationship between FDI and technological transactions. Additionally, marketization also shows a significant positive impact on technological innovation and technological transaction. With the enhancement of environmental regulation intensity, the production sector must quickly accomplish technical upgrades to achieve environmental compliance, which requires the production sector to enter the technology market to seek external technical support. Moreover, urbanization provides convenience for this relationship. With the deepening of urbanization and the acceleration of marketization, regions will attract more innovative resources, reduce the production costs of various departments, and improve the profits of the production sector. Moreover, urbanization can further optimize the allocation of R&D resources and gather a large number of researchers, which would improve the innovation ability of the region as a whole and promote the technological innovation of the region.

Thirdly, the heterogeneity study finds that the positive impacts of FDI on technological innovation through technology transactions and the moderating effect of environmental regulation exist only in eastern and western China. However, there are significant differences between the East and the West. In the eastern region, the marketization process is the main way to promote the above effect. Due to the developed economy in the eastern region, the marketization process there is deeper than that in other regions. Therefore, market forces play a decisive role in the resource allocation of the eastern region, so as to realize the agglomeration of R&D resources and profits. However, economic development of the western region lags behind that of the eastern region and its development is mainly realized through the process of urbanization, which realizes the agglomeration of regional innovation resources and the improvement of profitability. For the central region, its development level is between the east and the west, and it has completed the urbanization and enjoyed the development brought by urbanization. At the same time, due to the lag in market-oriented construction, the market's ability to optimize the allocation of resources does not appear in the region, therefore the technology spillover effect and innovation effect of FDI are not obvious in the central region.

**Author Contributions:** Conceptualization, M.Z. and Y.X.; methodology, M.Z. and Y.X.; software, M.Z.; investigation, L.L. and F.Z.; data curation, L.L. and F.Z.; writing—original draft preparation, M.Z.; writing—review and editing, Y.X. All authors have read and agreed to the published version of the manuscript.

**Funding:** This research was funded by Scientific research project of education department of Hunan province, grant number 20C0594.

**Institutional Review Board Statement:** The study did not involve humans or animals.

**Informed Consent Statement:** The study did not involve humans.

**Conflicts of Interest:** We declare that we have no financial and personal relationship with other people or organizations that can inappropriately influence our work, there is no professional or other personal interest of any nature or kind in any product, service and company that could be constructed as influencing the position presented in, or the review of, the manuscript entitled.

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
