# Peer review of "How Does Environmental Regulation Affect the Relationship between FDI and Technological Innovation: From the Perspective of Technology Transactions"

_processes, doi:10.3390/pr9081264_

Round 1

Reviewer 1 Report

This paper analyses the relationship between environmental regulation, FDI and innovation for provincial level data for China from 2005 to 2019. The paper could be improved by consideration of the following points:

  1. The sub-period analysis in table 4 shows that while FDI is significant in both time periods the marginal effect is larger in the second period. Could you consider the possibility of structural breaks in your analysis and add diagnostic tests as relevant?
  2. There are some recent papers that have explored the topic area that you could add to place your discussion in broader context, perhaps consider discussing Hao et al (2020) and Wang and Luo (2020).

References

Hao, Y., Wu, Y., Wu, H. and Ren, S. (2020), How do FDI and technical innovation affect environmental quality? Evidence from China, Environmental Science and Pollution Research 27, 7835–7850.

Wang, X. and Luo, Y. (2020), Has technological innovation capability addressed environmental pollution from the dual perspective of FDI quantity and quality? Evidence from China, Journal of Cleaner Production 258, 120941

Author Response

Respone to Reviewer #1

The authors thank for these helpful comments by reviewer sincerely. We revised our manuscript according to the comments above. The reviewer could check a new manuscript with all revisions uploaded by the authors in the system. All details of revision according to comments above are as follows. We also improve language in our manuscript according to comments by reviewer.

  1. The sub-period analysis in table 4 shows that while FDI is significant in both time periods the marginal effect is larger in the second period. Could you consider the possibility of structural breaks in your analysis and add diagnostic tests as relevant?

Thanks for this comments by reviewer. Table 4 is employed by the authors to prove that the results in table 3 are robust and credible. Sub-periods (2005-2012, 2011-2019) are chosen at random. The authors think that the different marginal effects (2.82** and 4.38**) can not strongly support the possibility of structural breaks. Hence, the authors do not provide relevant analysis and add diagnostic tests.

  1. There are some recent papers that have explored the topic area that you could add to place your discussion in broader context, perhaps consider discussing Hao et al (2020) and Wang and Luo (2020).

References

Hao, Y., Wu, Y., Wu, H. and Ren, S. (2020), How do FDI and technical innovation affect environmental quality? Evidence from China, Environmental Science and Pollution Research 27, 7835–7850.

Wang, X. and Luo, Y. (2020), Has technological innovation capability addressed environmental pollution from the dual perspective of FDI quantity and quality? Evidence from China, Journal of Cleaner Production 258, 120941

Thanks for this comments by reviewer. The references above actually provide some new evidences for us to support our thesis in the manuscript. In the new namuscript, the authors have cited these two articles as supportive conclusions.

Orginal manuscript

FDI has had a crucial impact on the environment of the host country. Due to the lag in the environmental system of the host country, FDI often exacerbates environmental pollution (Tong et al., 2021). And the lagging institutional environment of host countries makes TNCs regard developing countries as "Pollution Haven"(Pavlovic et al., 2021). As the Chinese government attaches greater importance to environmental protection, intensity of environmental regulation in China is on the rise. The Chinese governments tend to “screen” high-quality FDIs for greener economy benefits and sustainable development through strict environmental regulations (Qiu et al., 2021). High-quality FDIs concerned with advanced technologies could boost China’s energy efficiency and total factor productivity, and reduce pollution emissions (Pan et al., 2020a). Domestic firms and industries can make further innovation on the basis of absorbing advanced technology. There are not only positive impacts of environmental regulation on the quality of FDI but also on the innovation of domestic industries. For domestic firms, strict environmental regulation can trigger firms’ innovation that offset the innovation costs and promote competitiveness (Porter and Van Der Linde, 1995). ( from line 160 to line180)

Revised manuscript

FDI has had a crucial impact on the environment of the host country. Due to the lag in the environmental system of the host country, FDI often exacerbates environmental pollution (Tong et al., 2021). And the lagging institutional environment of host countries makes TNCs regard developing countries as "Pollution Haven"(Pavlovic et al., 2021). As the Chinese government attaches greater importance to environmental protection, intensity of environmental regulation in China is on the rise. The Chinese governments tend to “screen” high-quality FDIs for greener economy benefits and sustainable development through strict environmental regulations (Qiu et al., 2021). High-quality FDIs concerned with advanced technologies could boost China’s energy efficiency and total factor productivity, and reduce pollution emissions (Pan et al., 2020a). Wang and Luo (2020) found that when the level of FDI is low, the capacity for scientific and technological innovation aggravates environmental pollution levels, while when the level of FDI crosses a higher threshold, the capacity for scientific and technological innovation improves environmental quality. Hao et al. (2020) also confirm the existence of the "pollution halo hypothesis" that technological innovation can reduce the pollutant emissions. Domestic firms and industries can make further innovation on the basis of absorbing advanced technology. There are not only positive impacts of environmental regulation on the quality of FDI but also on the innovation of domestic industries. For domestic firms, strict environmental regulation can trigger firms’ innovation that offset the innovation costs and promote competitiveness (Porter and Van Der Linde, 1995). ( from line 160 to line180)

Reviewer 2 Report

This article analyzes the causal relationship between FDI and technological innovation in Chinese industry, considering the importance of environmental regulation in this relationship and its intensity. The analysis is addressed for the industrial sector of 30 provinces and regions of China, using various statistical sources (almost all official), and based on a correctly designed model, in our opinion, to meet the objectives of the research. Both the design of the model and its robustness and the coherence of the results obtained, allow the authors to reach conclusions that, also in our criteria, are sufficiently supported, and that are relevant to the field of study in which the investigation is framed.
For all the above, it is considered that the article meets the minimum requirements to be published in the journal, but providing the formal specifications of the journal and considering the possibility of incorporating some of the suggestions made below.
In the first place, the article has a more than sufficient review of the literature, but it is missing, within this, some works that with a similar methodology have addressed the issue in other regions or countries, with similar or different results to those obtained in the research.
Second, it also seems relevant, in the opinion of the reviewer of the article, to indicate whether there are studies or research results that have addressed the relationship between FDI and technological innovation in other non-manufacturing productive sectors, namely, in agricultural activities or in the third sector. In this sense, we consider that it would be a significant contribution of research to be able to contrast, albeit simply, the existence of differences that may exist along the different productive sectors in the effects that FDI produces on technological innovation, especially since environmental regulation of the productive sectors can be significantly different.
Third, we consider that section 3.1 should be expanded, being more explicit about the type of statistical information used. By way of example, the article speaks at all times of “industrial sectors”, but does not clarify what type of activities it refers to, or if figures have been used for the whole sector that includes all manufacturing activities. Particularly, we would like to know if investment in forests certification is taken into account when authors tal about investment in environmental governance.
Finally, the fragmentation of the analysis carried out by groups of regions distinguishes between eastern, western and central regions. The authors should justify the reason for this grouping. In any case, wouldn't a grouping of regions by income or development levels be more interesting for the analysis? This would also mean an additional contribution, depending on the results obtained, which would link FDI, Innovation and regional economic development.

Anyway, we congratulate authors for the research.

Author Response

Respone to Reviewer #2

The authors thank for these helpful comments by reviewer sincerely. We revised our manuscript according to the comments above. The reviewer could check a new manuscript with all revisions uploaded by the authors in the system. All details of revision according to comments above are as follows.

  1. In the first place, the article has a more than sufficient review of the literature, but it is missing, within this, some works that with a similar methodology have addressed the issue in other regions or countries, with similar or different results to those obtained in the research.

We supplement some literature review about this kind of study in other regions and countries. These studies also present different results. The supplementary literature review as follows (from line 124 to line 130, and from line 177 to line 182 in new manuscript):

Haug and Ucal (2019) presents new evidence that FDI has no statistically significant impact on environmental performance in Turkey. Results by Bakhsh et al. (2017) confirm that, in Pakistan, FDI can increase economic level that high economic level leads to lower pollutant emissions. Rafique et al. (2020) exmine the relationship between FDI and technological innovation in BRICs countries and find that no positive impact of FDI on technological innovation. (from line 124 to line 130)

Wang and Luo (2020) found that when the level of FDI is low, the capacity for scientific and technological innovation aggravates environmental pollution levels, while when the level of FDI crosses a higher threshold, the capacity for scientific and technological innovation improves environmental quality. Hao et al. (2020) also confirm the existence of the "pollution halo hypothesis" that technological innovation can reduce the pollutant emissions. (from line 177 to line 182)

  1. Second, it also seems relevant, in the opinion of the reviewer of the article, to indicate whether there are studies or research results that have addressed the relationship between FDI and technological innovation in other non-manufacturing productive sectors, namely, in agricultural activities or in the third sector. In this sense, we consider that it would be a significant contribution of research to be able to contrast, albeit simply, the existence of differences that may exist along the different productive sectors in the effects that FDI produces on technological innovation, especially since environmental regulation of the productive sectors can be significantly different.

The author team think that this is a good comments. Unfortunately, there are two reasons that this study can not present comparative results of the relationship between FDI and technological innovation in other non-manufacturing productive sectors, namely, in agricultural activities or in the third sector: (1) the official data on the third sector in China is from 2016 to 2019, which does not match the period of this article; (2) official reports and database do not provide the data of FDI in agricultural industry because of agricultural protection.

  1. Third, we consider that section 3.1 should be expanded, being more explicit about the type of statistical information used. By way of example, the article speaks at all times of “industrial sectors”, but does not clarify what type of activities it refers to, or if figures have been used for the whole sector that includes all manufacturing activities. Particularly, we would like to know if investment in forests certification is taken into account when authors tal about investment in environmental governance.

Thanks for the comment.   According to reviewer, we clarify the meaning of “industrial sectors” in this paper and investment in environmental governance. The revised details are as follows.

Orginal manuscript

3.1. Sample and data

In this study, industrial sectors of 30 provinces and regions in China are selected as research samples. Tibet, Hong Kong, Macau, and Taiwan are excluded in this study as data are not available. Innovation data come from the Yearbook of Industrial Science and Technology Statistics. Environmental regulation data come from the Yearbook of China's Environmental Statistics and the environmental statistics bulletin of various provinces and cities. Industry related data come from the Yearbook of China's Industrial Statistics, and data of marketization and urbanization come from China City Statistical Yearbook. Interpolation method is used to supplement the missing data. At the same time, this study uses the 1990 price index as the base period to deflate economic indicators to eliminate the impact of price changes on the results.

3.2. Variables

FDI, measured by amount of FDI received by industrial sectors each year (Yang and Li, 2019), is an independent variable in this study. Technological innovation is employed as dependent variable which is measured by the number of patent of industrial sectors (Ren et al., 2020). Technology transaction is the mediating variable and measured by volume of technology transaction (Goh, 2005). Environmental regulation is employed as the moderating variable and measured by investment in environmental governance according to Yuan and Xiang (2018). We also choose R&D capability measured by full-time equivalent of R&D (Zhang et al., 2019), inventory (Lee et al., 2015), profit (Douglas and Prentice, 2019), fixed investment (Skuras et al., 2008), size measured by number of employee (Ranasinghe, 2017), urbanization (Tu et al., 2018), and marketization (Zeng et al., 2021) as controls in this study.

Revised manuscript

3.1. Sample and data

In this study, industrial sectors of 30 provinces and regions in China are selected as research samples. Tibet, Hong Kong, Macau, and Taiwan are excluded in this study as data are not available. Industrial sectors, in this study, includes manufacturing industry, mineral mining industry, and energy production industry. Innovation data come from the Yearbook of Industrial Science and Technology Statistics. Environmental regulation data come from the Yearbook of China's Environmental Statistics and the environmental statistics bulletin of various provinces and cities. Industry related data come from the Yearbook of China's Industrial Statistics, and data of marketization and urbanization come from China City Statistical Yearbook. Interpolation method is used to supplement the missing data. At the same time, this study uses the 1990 price index as the base period to deflate economic indicators to eliminate the impact of price changes on the results.(from line 230 to 231 in new manuscript)

3.2. Variables

FDI, measured by amount of FDI received by industrial sectors each year (Yang and Li, 2019), is an independent variable in this study. Technological innovation is employed as dependent variable which is measured by the number of patent of industrial sectors (Ren et al., 2020). Technology transaction is the mediating variable and measured by volume of technology transaction (Goh, 2005). Environmental regulation is employed as the moderating variable and measured by investment in environmental governance according to Yuan and Xiang (2018). In this study, investment in environmental governance means expenditure from industrial sectors for reducing pollution from waste water, waste gas and solid waste produced by industrial sectors.We also choose R&D capability measured by full-time equivalent of R&D (Zhang et al., 2019), inventory (Lee et al., 2015), profit (Douglas and Prentice, 2019), fixed investment (Skuras et al., 2008), size measured by number of employee (Ranasinghe, 2017), urbanization (Tu et al., 2018), and marketization (Zeng et al., 2021) as controls in this study. (from line 246 to 248 in new manuscript)

  1. Finally, the fragmentation of the analysis carried out by groups of regions distinguishes between eastern, western and central regions. The authors should justify the reason for this grouping. In any case, wouldn't a grouping of regions by income or development levels be more interesting for the analysis? This would also mean an additional contribution, depending on the results obtained, which would link FDI, Innovation and regional economic development.

Thanks for the comment. Actually, we divide these samples into three regions based on economics development (GDP). According to reviewer, we supplement the expression to explain why we divide these samples into three regions.

Orginal manuscript

4.5. Heterogeneity analysis

Due to the huge development differences across various regions in China, this study has divided samples into various provinces and cities in China. Based on previous research management, this study divides China into three regions which include Eastern China, Central China, and Western China in which relationships among regional FDI, innovation, technology transaction, and environmental regulation are explored.

Revised manuscript

4.5. Heterogeneity analysis

Due to the huge development differences in economic development and geographic position across various regions in China, this study has divided samples into various provinces and cities in China. Based on previous research byYuan and Xiang (2018) and Yang and Li (2019), this study divides China into three regions which include Eastern China, Central China, and Western China in which relationships among regional FDI, innovation, technology transaction, and environmental regulation are explored. Eastern China includes Beijing, Tianjin, Hebei, Liaoning, Shanghai, Jiangsu, Zhejiang,Fujian, Shandong, Guangdong Guangxi and Hainan. Central  China includes Shanxi, Henan, Inner Mongolia, Anhui, Hubei, Hunan, Jiangxi, Jilin, and Heilongjiang. Western China is Shaanxi, Gansu, Ningxia, Qinghai, Xijiang, Chongqing, Sichuan, Guizhou and Yunnan. (from line 343 to 352 in new manuscript)